# Artificial Intelligence in Exercise Prescription in Palliative Care: Perceptions and Ethical Issues

**DOI:** 10.3390/healthcare13222987

**Published:** 2025-11-20

**Authors:** Daniela Oliveira, Francisca Rego, Rui Nunes

**Affiliations:** Faculty of Medicine, University of Porto, 4200-319 Porto, Portugal

**Keywords:** palliative care, artificial intelligence, exercise therapy, ethics, rehabilitation

## Abstract

**Highlights:**

**What are the main findings?**
Portuguese healthcare professionals expressed moderately positive perceptions of AI for exercise prescription in palliative care, with stronger agreement on ethical principles than on technical reliability.Greater knowledge of AI was associated with more favourable attitudes and fewer ethical concerns.

**What are the implications of the main findings?**
Responsible implementation of AI in palliative care requires structured training and ethical governance.AI should play a complementary role to human clinical judgement, safeguarding dignity, autonomy, and the humanization of care.

**Abstract:**

**Background/Objectives**: The integration of artificial intelligence (AI) in healthcare has been progressively expanding, with growing interest in its potential application in palliative care, particularly in exercise prescription. However, there is limited scientific evidence addressing professionals’ perceptions of this use. This study aimed to explore Portuguese healthcare professionals’ perceptions of AI in exercise prescription for palliative care and to identify related ethical implications and training needs. **Methods**: A cross-sectional, descriptive, and analytical study was conducted using an online questionnaire applied to health professionals. Sociodemographic data, knowledge, and perceptions of AI in exercise prescription in palliative care were collected. Descriptive and inferential statistical analyses were performed, including Mann–Whitney and Spearman correlation tests. **Results**: The sample consisted mainly of young female professionals with backgrounds in physiotherapy. Most participants reported little knowledge and experience with AI in clinical practice but expressed a strong interest in learning. Perceptions regarding the usefulness of AI were neutral to slightly positive, particularly concerning quality of life and personalization of interventions. Ethical aspects were viewed positively, especially the complementarity of AI to human clinical judgement, transparency, and explicability. Spearman correlations indicated that greater AI literacy and longer clinical experience were associated with more positive perceptions. **Conclusions**: The findings highlight the importance of developing structured training programmes that integrate practical, ethical, and technical components for the safe and responsible use of AI in palliative care. Ethical guidelines are proposed to ensure the humanization of care and the preservation of patient autonomy when implementing AI in exercise prescription.

## 1. Introduction

Modern medicine has been evolving towards an increasingly person-centred approach, something that is particularly evident when it comes to care during periods of greatest frailty or even at the end of life. Palliative care, as structured by Cicely Saunders in the 1960s, emerges as a deeply ethical and humanised response to the complex needs of people with incurable and progressive diseases that severely affect their quality of life [1,2,3]. More than just a set of practices, this care is now recognised as an essential human right [3,4], based on a comprehensive intervention that encompasses not only the body, but also the emotional, social, and spiritual well-being of the person—always with a focus on alleviating suffering, maintaining autonomy, and protecting dignity [5,6].

The development of palliative care in Portugal was relatively late compared to countries such as the United Kingdom, the United States and Canada. Until the 1990s, end-of-life care was provided in a fragmented manner, without formal recognition as a specialised area. Pioneering initiatives emerged in specialised hospitals such as the Portuguese Institute of Oncology in Porto (which in 1995 created the first palliative care service in Portugal, at the instigation of Prof. Ferraz Gonçalves) and through the actions of volunteers and non-profit organisations such as the Portuguese Association for Palliative Care (APCP), founded in 1995, which played a key role in promoting research and training in this area [7,8]. Currently in Portugal, palliative care is organised through the National Palliative Care Network (RNCP), formally established by Decree-Law No. 52/2012. This network, integrated into the National Health Service (SNS), aims to guarantee universal, equitable, and humane access to specialised care that alleviates suffering and promotes quality of life for people with chronic and progressive illnesses. The RNCP functions in coordination with the National Network for Integrated Continuing Care (RNCCI) and includes hospital, home, and community-based teams that adapt the level of intervention according to each patient’s needs [9]. Although the development of palliative care in Portugal occurred later than some other countries, initiatives led by the Portuguese Association for Palliative Care (APCP) and the creation of the first specialised units at the Portuguese Institute of Oncology in Porto were crucial milestones for national expansion. According to recent national assessments, the country has made significant progress in expanding specialised units and home-care teams, yet coverage remains uneven, particularly in rural and inland regions [7,8,10]. Nevertheless, regional asymmetries and limited professional training remain persistent challenges, reinforcing the importance of complementary interventions that enhance the quality of life and autonomy of patients receiving palliative care [7,8,9,10,11].

Rehabilitation and exercise prescription have increasingly attracted scientific attention as key non-pharmacological strategies in palliative care to improve quality of life. Evidence shows that structured exercise programmes can reduce fatigue, pain, and functional decline while promoting comfort, well-being, and residual functionality, even in advanced disease stages [12,13,14,15]. However, their systematic implementation remains difficult due to the heterogeneity of clinical conditions and the unpredictable evolution of symptoms. These limitations highlight the potential of artificial intelligence (AI) as an innovative tool to support personalised exercise prescription, enabling continuous adaptation of therapeutic programmes and providing clinical decision support in complex and changing scenarios [16,17,18].

AI-based systems trained with physiological, functional, and symptom-monitoring data can offer dynamic and individualised recommendations that evolve with the patient’s condition, supporting professionals in maintaining safety, autonomy, and proportionality of effort [19,20,21]. The literature has shown encouraging results, particularly regarding the early identification of signs of functional decline, the prediction of needs, and the management of symptoms [19,20,22]. This specificity explains the relevance of focusing on AI-supported exercise rather than on other interventions [13,16,17,18,22].

Although this integration remains in its early stages, growing scientific interest demonstrates a clear momentum toward applying AI in palliative contexts. Recent bibliometric evidence confirms that research on artificial intelligence in palliative care has expanded significantly in recent years, with 246 publications identified up to early 2024 and a marked growth between 2020 and 2024—reflecting the rapid but still emerging nature of this field (see Figure 1) [23].

However, the expansion of AI research in palliative care has also intensified ethical and practical debates about how these tools should be implemented in highly sensitive contexts. One of the main concerns relates to the lack of transparency of algorithms and the difficulty in clearly explaining how certain decisions are made by automated systems. This opacity can undermine the confidence of professionals and patients [24,25,26]. Furthermore, fundamental concepts such as autonomy and informed consent may be compromised when technology becomes too difficult or inaccessible for users [27]. Added to this is the fact that many healthcare professionals do not feel prepared, either in terms of digital literacy or ethical training, to integrate these tools in a safe and humanised manner [28,29].

Globally, several international organisations—such as the WHO, UNESCO and the OECD—have been publishing guidelines on the ethical use of AI in healthcare, based on principles such as beneficence, non-maleficence, justice and explainability [30,31,32]. However, these guidelines tend to be quite general and do not fully reflect the specificities of palliative care, where the goal is not to prolong life, but to ensure the best possible quality of life in the time remaining [13,26].

The scarcity of studies that gather the opinions and concerns of healthcare professionals about the use of AI in palliative care, particularly regarding prescribing exercise, still represents a gap in the literature. There is also a lack of practical guidelines, based on solid ethics, to help healthcare professionals make informed decisions about which tools to use, how to apply them, and how to assess their real impacts. This study therefore seeks to contribute to this debate by exploring healthcare professionals’ perceptions of the use of AI in prescribing therapeutic exercise in palliative care and proposing ethical guidelines that can promote more effective clinical practice—but without ever losing sight of the values of dignity, autonomy and humanisation of care.

## 2. Materials and Methods

### 2.1. Type of Study

A quantitative, descriptive, cross-sectional study aimed at assessing the perceptions, knowledge, and needs of healthcare professionals in Portugal regarding the use of artificial intelligence (AI) in prescribing therapeutic exercises in palliative care (PC). The quantitative approach allowed us to capture patterns, attitudes, and challenges in the clinical integration of artificial intelligence in palliative care.

### 2.2. Population, Sample and Context

Target population: Professionals working directly in PC, focusing on the prescription/monitoring of therapeutic exercise. Convenience sample, involving physiotherapists, doctors, nurses (rehabilitation) and other professionals with experience/interest in AI applied to PC, through an online questionnaire.

Setting: The study was conducted at the Faculty of Medicine of the University of Porto (FMUP), with institutional infrastructure and support, following the institution’s scientific and ethical standards.

Inclusion criteria:Professionals (physiotherapists, doctors, rehabilitation nurses and others) who prescribe or monitor therapeutic exercise in PC, including physical rehabilitation and quality of life interventions in a palliative context;Any level of experience with AI (includes only interest in learning/applying AI);Direct involvement in PC (hospital, home or community) with roles in prescribing/monitoring rehabilitation interventions;Tacit consent via progress in the questionnaire after reading the objectives/terms.

Exclusion criteria:

Professionals who did not work in palliative care or who failed to complete the questionnaire were excluded.

### 2.3. Data Collection Instrument 

Data were collected using a structured online questionnaire specifically designed for this study, based on bioethical principles (autonomy, beneficence, non-maleficence, justice) and transparency/clinical supervision/humanisation (WHO/UNESCO/Floridi references) [24,30,31].

The instrument consisted of 27 questions divided into four thematic dimensions: (I)Demographic and professional data—eight multiple-choice items characterising age, gender, professional category, region, academic background, years of experience in palliative care, training in the field, and previous contact with AI.(II)Perceptions and evaluations of AI—five items assessing knowledge, perceived usefulness, confidence in AI-generated recommendations, and expected clinical impact, using a five-point Likert scale (from “strongly disagree” to “strongly agree”, “very low” to “very high”, “very negative” to “very positive”, “ineffective” to “highly effective”).(III)Ethical dimension—nine questions exploring ethical concerns such as autonomy, privacy, transparency, and explainability, as well as one multi-response item addressing moral priorities in AI-supported care (e.g., balancing human judgement and algorithmic recommendations).(IV)Training and needs—five questions assessing the perceived need for education on AI, preferred content areas (ethics, algorithmic understanding, practical applications), and barriers to implementation, using Likert and multiple-choice formats.

The questionnaire was designed to ensure clarity and face validity, following methodological recommendations for health-technology acceptance studies [28,33]. This structure followed methodological standards used in recent studies exploring healthcare professionals’ perceptions of AI [27,32], ensuring comparability across contexts while adapting items to the palliative care setting.

The questionnaire underwent expert review by two specialists in ethics and palliative care to evaluate its clarity, relevance, and adequacy. As both evaluations were concordant and no modifications were suggested, full consensus was considered achieved.

### 2.4. Procedures

Participation in the study was entirely voluntary and anonymous, and no financial or material incentives were offered. Before accessing the questionnaire, participants were informed of the study’s objectives, the voluntary nature of participation, and data protection procedures. No identifiable personal data (such as name, email, IP address, or institution) were collected. Tacit informed consent was provided by continuing to the questionnaire after reading these terms.

The questionnaire was administered online and disseminated by email, remaining open for approximately three months (March–May 2025).

The study was conducted in accordance with the Declaration of Helsinki and the General Data Protection Regulation (GDPR) and was approved by the Ethics Committee of the Faculty of Medicine of the University of Porto (protocol 318/CEFMUP/2025). Data were securely stored on an encrypted database accessible only to the principal investigator, ensuring full confidentiality and compliance with ethical standards.

A total of 72 valid responses were obtained; only fully submitted questionnaires were considered, as the form required completion of all items before submission. The final sample included 34 physiotherapists, 18 rehabilitation nurses, 12 physicians, and 8 occupational therapists, all directly involved in palliative care and rehabilitation.

### 2.5. Data Analysis

Prior to statistical analysis, the dataset was verified for completeness, as only fully submitted questionnaires were considered valid. Since the online form required completion of all items before submission, no missing data or duplicates were present.

Descriptive statistics (frequencies, percentages, means, and standard deviations) were used to characterise the sample and summarise responses. Given the ordinal nature of the Likert-scale variables and the non-normal distribution of data, non-parametric tests were applied: Spearman’s rank correlation was used to examine associations between variables, and Mann–Whitney U tests were used to compare independent groups.

The rationale for the correlation analysis was to explore possible relationships between professional experience, AI literacy, and perceptions of artificial intelligence in palliative care. As the variables were ordinal (Likert scale and grouped experience levels), Spearman’s correlation (ρ) was applied to identify monotonic associations. The length of professional experience in palliative care was treated as an ordinal variable (three ordered categories: <1 year, 1–5 years, >5 years), allowing the detection of potential experience-related trends without assuming linearity.

Given the exploratory nature of the study, no correction for multiple comparisons was applied, as the aim was descriptive—to identify patterns that may guide future hypothesis-driven research. Responses marked as “I don’t have an opinion” were treated as non-evaluable and excluded from ordinal statistical analyses. These options appeared in items addressing the perceived efficacy of AI and several ethical questions (e.g., transparency, review of AI recommendations, and patient information). Their exclusion was necessary to preserve the ordinal nature of the data and the validity of non-parametric tests.

For comparative analysis, participants were divided into two groups according to whether they possessed specialised training in palliative care. These comparisons aimed to explore potential differences in perceptions, ethical considerations, and readiness to integrate AI in exercise prescription between professionals with and without palliative care specialisation.

All analyses were conducted using IBM SPSS Statistics v26.0 by the principal investigator.

## 3. Results

A total of 72 healthcare professionals participated in the study, predominantly female (61.1%), with ages ranging from 31 to 40 years (54.2%). The sample included physiotherapists (n = 52), nurses (n = 9), physicians (n = 5) and occupational therapists (n = 6). The predominance of female participants reflects the gender distribution typically observed in the Portuguese healthcare workforce, particularly in nursing and physiotherapy.

There was a strong concentration in the North (87.5%; n = 63). The geographical concentration of participants in the Northern region represents a limitation of this study, potentially restricting the generalisability of results to other Portuguese regions. This bias likely reflects the location of the research institution and the convenience-based recruitment strategy rather than differences in perceptions among professionals.

Training in physiotherapy predominated (72.2%; n = 52), followed by nursing (12.5%; n = 9), occupational therapy (8.3%; n = 6) and medicine (6.9%; n = 5). Regarding specific training in palliative care, 66.7% (n = 48) reported having no training in the area and 33.3% (n = 24) reported having continuous/specialised training.

Professional experience in palliative care was classified in three categories (<1 year, 1–5 years, and >5 years). This categorical format was intentionally adopted to distinguish between novice, intermediate, and experienced professionals, facilitating interpretation in this descriptive and exploratory study. Although no standardised intervals exist, this approach aligns with recommendations to consider different stages of clinical and technological experience when evaluating professional perceptions [28,29,33].

Overall, 36.1% (n = 26) of participants reported 1–5 years of experience, 30.6% (n = 22) had more than 5 years, and 16.7% (n = 12) had less than 1 year of professional experience in palliative care (see Table 1).

Experience with artificial intelligence was assessed through a four-level categorical item: “no practical experience”, “initial experience (occasional or limited use)”, “moderate experience (regular use in some clinical situations)”, and “advanced experience (frequent and integrated use in clinical practice)”. The category “moderate experience” referred to participants who reported regular but not systematic use of AI in specific clinical contexts.

Practical experience with AI was mostly moderate (51.4%; n = 37). Despite this, interest in learning how to use AI in exercise prescription was high: 56.9% (n = 41) reported high interest and 38.9% (n =28) moderate interest, with residual cases of low interest (4.2%; n = 3). Regarding knowledge of AI in prescription, 50.0% (n = 36) reported little knowledge, 30.6% (n = 22) average knowledge, 4.2% (n = 3) good knowledge, and 15.3% (n = 11) no knowledge. Overall, the results point to a high willingness to undergo training and a clear training gap that needs to be addressed (See Table 2.).

The analysis of healthcare professionals’ perceptions of artificial intelligence (AI) in palliative care revealed different trends depending on the dimension assessed (see Table 3). Regarding the usefulness and reliability of AI, question Q10, on the contribution of AI to improving patients’ quality of life, had a mean of 3.1 (SD = 0.7; 25th and 75th percentiles = 3 and 4), showing a positive correlation between level of AI experience and perceived usefulness, indicating that greater familiarity with AI was associated with higher perceived usefulness.

In Q11, regarding the effectiveness of AI in personalising exercise programmes, the mean was 3.3 (SD = 0.6; 25th and 75th percentiles = 3 and 4), excluding 25% of participants who indicated “I don’t know”, suggesting a moderately positive perception. On the other hand, Q12, which assessed the impact of AI on mobility, pain relief, and quality of life, had a mean of 1.9 (SD = 0.7; 25th and 75th percentiles = 1 and 2), indicating a negative trend. Confidence in AI-generated recommendations (Q13) revealed a mean of 2.6 (SD = 1.0; 25th and 75th percentiles = 2 and 3), showing a slightly negative to moderate trend.

Regarding the ethical dimension, participants demonstrated a high level of agreement with key ethical principles. For Question 15, concerning the principle that AI should complement rather than replace human judgement, the mean score was 4.6 (SD = 0.6; 25th–75th percentiles = 4–5). For Question 16, addressing transparency in algorithmic decisions, the mean score was 4.5 (SD = 0.7; 25th–75th percentiles = 4–5).

The importance of algorithm explainability (Q17) revealed an average of 4.2 (SD = 0.8; 25th and 75th percentiles = 4 and 5). In contrast, the perception that AI could compromise the empathic relationship (Q22) had a mean of 1.9 (SD = 0.9; 25th and 75th percentiles = 1 and 3), indicating that professionals consider any negative impact on relational empathy to be unlikely.

Regarding training and needs, Q26, on the potential of AI to reduce inequalities in access to palliative care, had a mean score of 3.8 (SD = 0.9; 25th and 75th percentiles = 3 and 5), showing a positive trend. The acceptance of AI among colleagues (Q27) scored an average of 3.9 (SD = 1.0; 25th and 75th percentiles = 3 and 5), suggesting a relatively high perception of institutional and professional acceptance.

In summary, the results show that professionals expressed reservations regarding the direct clinical effectiveness and reliability of algorithmic recommendations. Nonetheless, most participants agreed that AI should complement rather than replace human judgement, highlighting the importance of transparency and explainability in algorithmic processes. Participants also rated potential contributions of AI to equity in access and institutional integration favourably. These findings suggest general acceptance of AI as a supportive tool within palliative care practice.

Figure 2 shows the most relevant ethical concerns related to the use of AI in palliative care, allowing multiple responses to be selected. The humanisation of care stands out as the main concern (86.1%; n = 62), followed by data privacy (75.0%; n = 54), patient autonomy (44.4%; n = 32) and algorithm explainability (19.4%; n = 14). Together, these results highlight the priority given to preserving the therapeutic relationship, information security/confidentiality and self-determination, with transparency/explainability remaining a relevant ethical requirement for trust in AI-assisted clinical decision-making.

On categorical ethical issues, the majority advocated clinical supervision of AI recommendations: 54.2% considered that they should always be reviewed by a professional and 27.8% believed that review is necessary in critical situations; opposing positions were residual (12.5% responded “not necessary” and 5.6% responded “don’t know” (Q18)). Regarding information to patients/family members about the use of AI, responses were distributed between informing only if requested (34.7%), always informing (27.8%) and the team’s exclusive decision (33.3%), with 4.2% responding “don’t know” (Q19). In the event of a conflict between the AI recommendation and the patient’s decision, patient autonomy prevailed (50.0%), followed by joint team–patient decision (47.2%) (Q20). Regarding explainability and acceptance, moderate acceptance with simplified explanations predominated (43.1%), followed by total acceptance only with detailed explanations (38.9%); 5.6% considered it irrelevant and 12.5% said “don’t know” (Q21). Taken together, the results reinforce the centrality of human clinical judgement, the right to information, and explainability as pillars for trust and ethical legitimacy in the use of AI in PC (see Table 4).

Table 5 shows the correlations between length of experience in palliative care, level of knowledge about AI in prescribing therapeutic exercises, and the dimensions assessed in the questionnaire. Length of experience showed significant positive correlations with the perception that AI contributes to improving patients’ quality of life (Q10: ρ = 0.252; *p* < 0.05), confidence in AI-generated recommendations (Q13: ρ = 0.240; *p* < 0.05) and acceptance of AI among colleagues (Q27: ρ = 0.260; *p* < 0.05), indicating greater appreciation of the technology with experience. In contrast, it showed a significant negative correlation with the perception that AI may compromise the empathetic relationship with patients (Q22: ρ = −0.309; *p* < 0.01), suggesting that more experienced professionals perceive a lower risk of dehumanisation.

The level of knowledge about AI was positively correlated with several items in the utility and reliability dimension: AI’s contribution to quality of life (Q10: ρ = 0.373; *p* < 0.01); effectiveness in personalising exercise programmes (Q11: ρ = 0.594; *p* < 0.01); impact on mobility, pain relief and quality of life (Q12: ρ = 0.407; *p* < 0.01); and confidence in recommendations (Q13: ρ = 0.501; *p* < 0.01). This shows that higher knowledge levels were associated with more positive perceptions. In the ethical dimension, negative correlations were observed with the idea that AI should complement, not replace, human judgement (Q15: ρ = −0.251; *p* < 0.05) and with the perception of possible compromise of the empathic relationship (Q22: ρ = −0.269; *p* < 0.05), indicating that greater AI knowledge tended to correspond with fewer ethical concerns. Regarding the training/needs dimension, knowledge was positively associated with the perception of reduced inequalities in access to palliative care (Q26: ρ = 0.333; *p* < 0.01) and with acceptance among colleagues (Q27: ρ = 0.260; *p* < 0.05).

In summary, both experience and knowledge of AI showed significant associations with professionals’ perceptions. Technical knowledge was related to greater appreciation of AI’s usefulness and reliability, perceived clinical impact, and confidence in recommendations, while being inversely associated with certain ethical concerns. Professional experience, in turn, was positively associated with acceptance of AI and perception of clinical benefits, while being negatively correlated with the idea of risk to human relationships. Although the correlation coefficients were of low to moderate magnitude, they provide indicative patterns consistent with the exploratory and descriptive nature of the study, suggesting potential relational trends that warrant further investigation. These findings highlight the importance of promoting AI literacy as a strategy for safe, critical, and informed adoption in palliative care (see Table 5).

The comparison by specialisation in palliative care did not reveal statistically significant differences in any of the analysed dimensions (usefulness/reliability, ethics, or training/needs; all *p* > 0.05). Although descriptive means were slightly higher among professionals with formal PC specialisation for certain items, these differences were minimal. No meaningful group effects were detected, indicating that perceptions of AI were broadly consistent regardless of specialisation status (see Table 6).

Figure 3 shows the distribution of priority aspects in training on Artificial Intelligence (AI), as perceived by the healthcare professionals participating in the study (N = 72). It should be noted that participants were able to select more than one answer option simultaneously, which made it possible to identify multiple training areas considered relevant, ranging from practical skills to ethical dimensions and technical understanding of AI.

The most prominent aspect was practical training on the use of AI in rehabilitation, identified by 88.9% (n = 64) of participants, highlighting the value placed on the concrete applicability of the technology in clinical practice. In second place were the ethical aspects of AI use, highlighted by 68.1% (n = 49) of respondents, demonstrating concern about professional conduct and the ethical limits of working with AI.

Understanding the impacts on patient autonomy and dignity was considered a priority by 62.5% (n = 45) of participants, reflecting sensitivity to the preservation of users’ fundamental rights. Finally, understanding algorithms and how AI works was mentioned by 59.7% (n = 43) of professionals, revealing an interest in the explainability and technical functioning of the tool, albeit to a lesser extent than the other dimensions.

These data reinforce the importance of a training model that combines practical, ethical and digital literacy components, promoting the informed, critical and humanised use of AI in rehabilitation and palliative care contexts.

## 4. Discussion

### 4.1. Main Findings and Context

This study found that healthcare professionals generally hold positive but cautious perceptions of AI in palliative care, with limited practical experience but strong interest in training and ethical awareness. The progressive rise in publications on artificial intelligence in palliative care in recent years [23] reinforces the timeliness of this study, which contributes by exploring the perceptions and ethical challenges associated with its clinical integration. This discussion aims to critically examine the results obtained, based on the limited literature available on the application of artificial intelligence (AI) in exercise prescription in palliative care. Recognising this limitation, we resort to comparisons with studies conducted in other areas of health, such as neurological, orthopaedic, cardiac and respiratory rehabilitation, where AI has been integrated more robustly.

To our knowledge, this is the first empirical study to examine healthcare professionals’ ethical perceptions of AI-assisted exercise prescription specifically within palliative care.

### 4.2. Perceptions, Experience, and Knowledge

The predominance of young female participants in the sample may explain the higher representation of this demographic among those expressing favourable perceptions of AI, reflecting the gender and age distribution typical of rehabilitation and palliative care professions in Portugal rather than a true difference in attitudes between demographic groups. Like the present study, the participating population in Hoffman et al. [33] was also mostly young, female, and with an average of 10 years of clinical practice; these participants considered that AI could improve health services, recognising multiple opportunities for its application [33].

Regarding experience and knowledge, it was found that most have little experience in using AI in clinical practice and little knowledge about prescribing therapeutic exercises with AI, but a high interest in learning how to use these tools. This pattern converges with the literature that points to globally low levels of digital literacy in AI among health professionals [28].

Regarding the dimensions assessed, the results show neutral to slightly positive perceptions of AI’s contribution to quality of life and the personalisation of interventions (Q10–Q11), however with reservations regarding the direct impact in mobility/pain (Q12) and trust regarding recommendations (Q13). In contrast, the ethical dimension shows a strongly positive trend, with an appreciation of complementarity to human clinical judgement (Q15), transparency (Q16) and explainability (Q17); concern that AI compromises the empathic relationship (Q22) is lower, suggesting confidence in the possibility of reconciling technology and humanisation. In the training/needs dimension, two relevant perceptions stand out: that AI can contribute to reducing inequalities in access to healthcare and rehabilitation interventions in a palliative context (Q26) and that there is growing acceptance among peers (Q27). This set of results supports the need for structured and multidimensional training, integrating practical, ethical and technical components for safe, informed and responsible implementation. 

These results suggest that healthcare professionals value AI primarily as a complementary and ethically supervised resource, rather than a replacement for human decision-making. The emphasis placed on transparency, explainability, and equity reflects an understanding of AI as a tool that can enhance—but not redefine—palliative care practice, provided its implementation is guided by ongoing ethical reflection and adequate professional training.

### 4.3. Comparisons with International Literature

Although direct evidence in palliative care remains limited to date, studies in oncology and rehabilitation contexts have shown promising results with AI systems for early identification of functional decline, prediction of care needs, and adaptation of exercise programmes to patient capacity [13,22,34,35,36]. These findings support the potential translatability of such approaches to palliative care rehabilitation, where individualisation, safety, and symptom monitoring are essential [37]. The integration of these resources demonstrates that technology can significantly complement clinical practice without replacing the role of the professional, reinforcing the need for ethically guided and clinically supervised implementation in palliative rehabilitation.

The Spearman correlations presented in Table 5 suggest that specific AI literacy and clinical experience are associated with professionals’ perceptions of artificial intelligence in palliative care. Greater knowledge tended to align with more positive perceptions of usefulness and effectiveness, fewer ethical concerns, and a more favourable view of AI as an instrument of equity (Q26) and acceptance (Q27). Length of experience in PC was associated with higher appreciation of AI’s potential to improve quality of life (Q10), greater confidence in recommendations (Q13), and reduced concern about loss of empathy (Q22). Although the observed correlation coefficients were of low to moderate magnitude, these relationships indicate emerging trends consistent with the exploratory nature of the study. Taken together, the results underscore the relevance of promoting AI literacy among professionals to support safe, critical, and informed adoption in palliative care.

When compared with international benchmark studies, the Portuguese findings reveal a comparable pattern of cautious optimism and ethical awareness. In the systematic review by Sumner et al. (2023) [21], AI-assisted rehabilitation tools—particularly in neurological and musculoskeletal contexts—were associated with improved functional outcomes and adherence, though concerns regarding algorithmic transparency and clinician training persisted. Likewise, Vu et al. (2023) [22] reported that although predictive models demonstrate high accuracy in symptom detection and prognosis, their integration into practice remains limited by ethical concerns and contextual barriers. These trends mirror the perceptions observed in the present study, where professionals valued explainability and human supervision as prerequisites for adoption.

However, in contrast to higher-resource settings such as the United Kingdom and Canada—where digital readiness and multidisciplinary AI collaborations are more consolidated [7,29]—the Portuguese context still faces structural challenges, particularly the scarcity of AI-specific training. This disparity underscores the importance of implementation strategies and ethically grounded training programmes tailored to national realities.

Comparable research conducted in Germany, the United Kingdom, and Canada has likewise identified limited AI literacy and the absence of clear ethical frameworks as persistent barriers to clinical integration, despite generally positive attitudes toward technological innovation [23,28,33]. Together, these international parallels indicate that the challenges identified in Portugal are part of a broader global landscape, reinforcing the need for shared educational and policy initiatives that promote responsible, equitable, and human-centred AI adoption in healthcare.

### 4.4. Ethical and Practical Implications

In ethical-practical terms, the results are in line with the literature that maps AI ethics in healthcare: the priority given to explainability, transparency, fairness, and preservation of human clinical judgement is consistent with recent principles and mappings, while international guidelines reinforce the need for ethical governance, accountability, and informational proportionality in clinical relationships [6,25,30].

The data support the implementation of training programmes that combine applied training (selection/use of tools; critical reading of outputs), applied ethics modules (consent, proportional explainability, data governance) and clinical communication skills to preserve humanisation and autonomy. This training approach is consistent with the priorities identified by the sample (practical training, ethics, autonomy/dignity, understanding of algorithms).

Building on these findings, a set of ethical guidelines (see Table 7) was developed to support the safe and responsible integration of AI into exercise prescription for palliative care. Each ethical guideline presented below was formulated based on the results of this study and grounded in established ethical frameworks and empirical evidence. The selected references encompass complementary perspectives, including foundational bioethical principles such as autonomy, dignity, and justice; international governance documents that define ethical standards for AI in healthcare (e.g., WHO, UNESCO); and recent studies addressing practical challenges such as informed consent, transparency, fairness, and digital literacy. Collectively, these sources provide a coherent and evidence-based foundation for the proposed guidelines, ensuring their applicability to the ethical, safe, and humanised use of AI in exercise prescription within palliative care.

To illustrate the practical application of these ethical guidelines, consider the following hypothetical scenario.

In a conventional rehabilitation context, a physiotherapist prescribes a moderate-intensity exercise plan based on the patient’s observed fatigue and verbal feedback. In contrast, an AI-assisted system might recommend a higher exercise intensity based on predictive modelling of physiological data, such as heart rate variability or prior performance trends.

Although the AI-generated recommendation is evidence-based, the clinician must assess whether it aligns with the patient’s current comfort, emotional state, and palliative goals.

This comparison highlights the ethical importance of maintaining human oversight and respecting patient preferences. The clinician’s role remains essential in contextualising algorithmic outputs, ensuring that recommendations are proportional, compassionate, and adapted to each patient’s lived experience—principles that are central to palliative care.

### 4.5. Limitations and Future Directions

Convenience sampling, sample size, and cross-sectional/self-report nature limit generalisation and prevent causal inferences. Geographical concentration and the predominance of certain professional profiles may introduce selection bias. The scarcity of specific literature on AI in exercise prescription in PC reinforces the need for interpretative caution and replication in more heterogeneous samples. The development of longitudinal and interventional studies evaluating the impacts of AI on clinical and experiential outcomes (quality of life, symptoms, satisfaction), as well as implementation assessments (acceptance, safety, effectiveness) and explainability tests adapted to the PC rehabilitation context, is recommended.

## 5. Conclusions

This study provides empirical insight into healthcare professionals’ perceptions of artificial intelligence (AI) in exercise prescription within palliative care. Overall, participants demonstrated cautious optimism—recognising the potential of AI to enhance personalisation and equity of care, while emphasising the need for human oversight, transparency, and ethical governance.

The findings reveal that limited AI literacy, insufficient ethical guidance, and lack of context-specific training remain major barriers to implementation. These challenges underscore the importance of developing national training programmes and institutional frameworks that integrate technical, ethical, and communicational competencies.

Building on these results, the proposed ethical guidelines offer a preliminary framework for safe, equitable, and human-centred AI adoption in palliative rehabilitation. Future research should focus on testing these principles in practice through longitudinal and interventional studies that assess the real-world impacts of AI-assisted rehabilitation on quality of life, safety, and clinical outcomes in palliative care.

## Figures and Tables

**Figure 1 healthcare-13-02987-f001:**
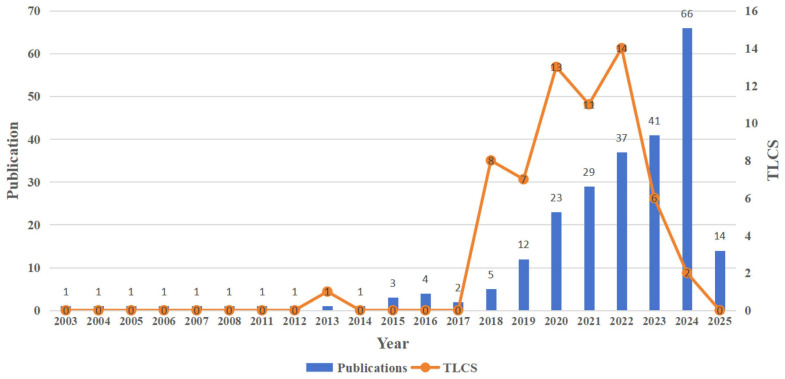
Annual publication trends on artificial intelligence in palliative care (2013–2024). Bars show yearly publication counts; the line shows Total Local Citation Score (TLCS). Source: Pan et al., 2025, Frontiers in Medicine, CC BY 4.0 [23].

**Figure 2 healthcare-13-02987-f002:**
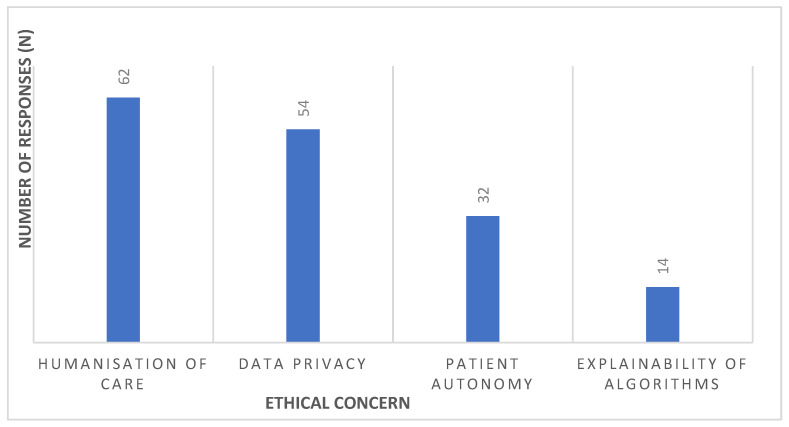
Most relevant ethical concerns in the use of AI in palliative care.

**Figure 3 healthcare-13-02987-f003:**
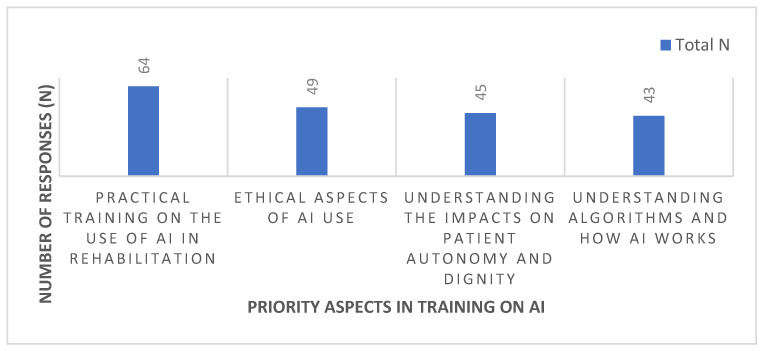
Priority aspects in AI training.

**Table 1 healthcare-13-02987-t001:** Socio-professional frequency variables.

	N	%
**Gender**		
Female	44	61.1
Male	28	38.9
**Age Gap**		
18–30 years old	16	22.2
31–40 years old	39	54.2
41–50 years old	13	18.1
51–60 years old	2	2.8
>60 years old	2	2.8
**Residency region/performance in Portugal**		
Centre	8	11.1
Lisbon & Tejo Valley	1	1.4
North	63	87.5
**Main academic qualifications**		
Nursing	9	12.5
Physiotherapy	52	72.2
Medicine	5	6.9
Occupational Therapy	6	8.3
**Specialisations or continuing education in palliative care**		
No	48	66.7
Yes	24	33.3
**Experience in palliative care**		
<6 months	12	16.7
6 months–1 year	12	16.7
1–5 years	26	36.1
>5 years	22	30.6

**Table 2 healthcare-13-02987-t002:** Frequency of AI experience level, interest in learning how to use AI, and knowledge about AI.

	N	%
**Q7_ Level of experience with artificial intelligence in clinical practice**		
No practical experience	14	19.4
Initial experience (occasional or limited use)	21	29.2
Moderate experience (in some clinical situations)	37	51.4
**Q8_ Interest in learning how to use AI in prescribing therapeutic exercises**		
Low interest	3	4.2
Moderate interest	28	38.9
High interest	41	56.9
**Q9_ Level of knowledge about AI in prescribing therapeutic exercises**		
None	11	15.3
Little	36	50.0
Moderate	22	30.6
Good	3	4.2

**Table 3 healthcare-13-02987-t003:** Frequency of Usefulness and Reliability of AI, Ethical Dimension in AI, and Training/Need for AI, depending on the presence of Specialisation in Palliative Care.

		N	Mean	SD	Min	Max	Percentiles	
							25	75
**Usefulness and reliability of AI**								
Q10_ AI can contribute to improving patients’ quality of life		72	3.1	0.7	1	4	3	4
Q11_ The effectiveness of AI in personalising exercise programmes		54	3.3	0.6	1	4	3	4
Q12_ AI can impact patients’ mobility, pain relief, and quality of life		72	1.9	0.7	1	3	1	2
Q13_ Trust in AI-generated recommendations in clinical practice		72	2.6	1.0	1	4	2	3
**Ethic Dimension**								
Q15_ AI should complement, not replace, human judgement in palliative care		72	4.6	0.6	1	5	4	5
Q16_ Transparency about AI-generated decisions is essential		72	4.5	0.7	1	5	4	5
Q17_ The explainability of algorithms is crucial for the trust of healthcare professionals		72	4.2	0.8	1	5	4	5
Q22_ The use of AI can compromise the empathetic relationship between healthcare professionals and patients		72	1.9	0.9	1	4	1	3
**Training and Needs for AI**								
Q26_ AI can help reduce inequalities in access to palliative care		72	3.8	0.9	2	5	3	5
Q27_ Assess AI acceptance among colleagues		72	3.9	1.0	1	5	3	5

Note. Values represent mean Likert-scale scores and standard deviations. Non-integer mean values reflect the averaging of ordinal responses across participants.

**Table 4 healthcare-13-02987-t004:** Frequency of questions on ethical issues.

	N	%
**Q18_ Considers it essential that all recommendations made by AI are always reviewed by a healthcare professional**		
Essential only in critical situations	20	27.8
Essential in all situations	39	54.2
Not necessary for general recommendations	9	12.5
Don’t know	4	5.6
**Q19_ Patients and family members should be explicitly informed about the use of AI in prescribing exercises**		
Yes, but only if requested by the patient	25	34.7
No, the decision rests solely with the medical team	24	33.3
Yes, always	20	27.8
Don’t Know	3	4.2
**Q20_ When there is a conflict between the AI recommendation and the patient’s decision, which should take priority**		
Patient autonomy	36	50.0
Joint decision between clinical team and patient	34	47.2
Don’t Know	2	2.8
**Q21_ The explainability of AI algorithms affects their acceptance as a clinical tool**		
Moderate acceptance with simplified explanations	31	43.1
Full acceptance only with detailed explanations	28	38.9
Not relevant to acceptance	4	5.6
Don’t Know	9	12.5

**Table 5 healthcare-13-02987-t005:** Correlation between length of experience in palliative care, level of knowledge about AI in prescribing therapeutic exercises, and the dimensions of AI usefulness and reliability, ethical dimension in AI, and training/need for AI.

	Level of Experience in Palliative Care	Level of Knowledge About AI in Exercise Prescription
**Usefulness and reliability of AI**		
Q10_ AI can contribute to improving patients’ quality of life	0.252 *	0.373 **
Q11_ The effectiveness of AI in personalising exercise programmes	0.219	0.594 **
Q12_ AI can impact patients’ mobility, pain relief, and quality of life	0.056	0.407 **
Q13_ Trust in AI-generated recommendations in clinical practice	0.240 *	0.501 **
**Ethic Dimension**		
Q15_ AI should complement, not replace, human judgement in palliative care	−0.143	−0.251 *
Q16_ Transparency about AI-generated decisions is essential	−0.046	−0.096
Q17_ The explainability of algorithms is crucial for the trust of healthcare professionals	−0.02	−0.103
Q22_ The use of AI can compromise the empathetic relationship between healthcare professionals and patients	−0.309 **	−0.269 *
**Training and Needs for AI**		
Q26_ AI can help reduce inequalities in access to palliative care	0.211	0.333 **
Q27_ Assess AI acceptance among colleagues	0.260 *	0.260 *

* The correlation is significant at the 0.05 level (two-tailed); ** the correlation is significant at the 0.01 level (two-tailed). Note: Spearman’s rank correlation coefficients (ρ) are reported.

**Table 6 healthcare-13-02987-t006:** Comparison of the Usefulness and Reliability of AI, Ethical Dimension in AI, and Training/Need for AI, depending on the presence of Specialisation in Palliative Care.

	Specialisations/Continuing Education in Palliative Care							
	No				Yes			
	N	Mean	SP	N	Mean	SP	Dif	*p*
**Usefulness and reliability of AI**								
Q10_ AI can contribute to improving patients’ quality of life	48	3.1	0.7	24	3.0	0.8	−0.1	0.754
Q11_ The effectiveness of AI in personalising exercise programmes	48	2.4	1.5	24	2.7	0.5	0.3	0.288
Q12_ AI can impact patients’ mobility, pain relief, and quality of life	48	1.9	0.6	24	2.0	0.8	0.1	0.657
Q13_ Trust in AI-generated recommendations in clinical practice	48	2.5	0.9	24	2.6	1.0	0.1	0.591
**Ethic Dimension**								
Q15_ AI should complement, not replace, human judgement in palliative care	48	4.7	0.4	24	4.4	0.9	−0.3	0.095
Q16_ Transparency about AI-generated decisions is essential	48	4.6	0.6	24	4.5	0.9	−0.1	0.989
Q17_ The explainability of algorithms is crucial for the trust of healthcare professionals	48	4.3	0.7	24	4.0	1.0	−0.3	0.292
Q22_ The use of AI can compromise the empathetic relationship between healthcare professionals and patients	48	1.9	0.9	24	2.0	0.9	0.1	0.621
**Training and Needs for AI**								
Q26_ AI can help reduce inequalities in access to palliative care	48	3.8	0.9	24	4.0	0.9	0.3	0.307
Q27_ Assess AI acceptance among colleagues	48	4.0	0.9	24	3.8	1.2	−0.2	0.469

**Table 7 healthcare-13-02987-t007:** Proposed ethical guidelines for the use of AI in exercise prescription in palliative care.

	Ethic Guideline	Scientific Justification
1	AI should complement—and never replace—human clinical judgement.	Clinical judgement ensures contextualised decisions. The absence of this can dehumanise care [24,30].
2	AI recommendations should be tailored to the patient’s preferences and limitations.	Patient autonomy is central to palliative care [5,6,13]
3	There should be constant clinical supervision of the exercises suggested by AI.	AI is a tool, not an autonomous agent; the absence of supervision compromises therapeutic proportionality [36]
4	The choice of technology should be based on validated and transparent evidence.	Explainability has increased trust in AI [24]
5	The informed consent of the patient and/or carer must be obtained.	Algorithmic opacity hinders understanding; it is necessary to adapt language and verify the digital literacy of those involved [25,27,28]
6	The algorithms used must respect the principles of fairness and equity in access.	Biased algorithms can perpetuate inequalities; implementation must be fair [30,38]
7	The use of AI should be transparent and shared with the entire interdisciplinary team.	Sharing promotes informed and collaborative clinical decisions, which are fundamental in palliative care [39,40]
8	The training of professionals in digital literacy and algorithmic ethics must be ongoing.	Lack of training is a barrier to the safe adoption of AI; teams need to be empowered to ensure humanised care [28,29]

Note: Table prepared by the authors based on data from this study and guidelines from the literature. The proposed guidelines are intended as a preliminary framework to inform clinical training and institutional policy development.

## Data Availability

The raw data supporting the conclusions of this article will be made available by the authors upon request.

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
