# Peer review of "Artificial Intelligence in Exercise Prescription in Palliative Care: Perceptions and Ethical Issues"

_healthcare, 2025, doi:10.3390/healthcare13222987_

Round 1
Reviewer 1 Report
Comments and Suggestions for Authors
The manuscript, Artificial Intelligence in Exercise Prescription in Palliative Care: Perceptions and Ethical Issues is written on interesting topic for the research fraternity. The main finding is responsible implementation of AI in palliative care requires structured training and ethical governance. Some suggestions needs to be added to improve quality:
- Introduction part needs the support of figures so that topic can be well understand , like no. of publications in the same field yearly.
- Materials and Methods section can be more detailed for the readers. Same studies covered by other researchers must be compared.
- The result section is well written , still it is oriented with its own case-study. Authors should compare Portuguese findings with existing benchmark datasets on AI in rehabilitation and palliative care to identify contextual barriers and best practices.
- Write role-play simulations presenting difference between AI-generated and non-AI generated; recommendations and patient preferences.
- The conclusion must be re-written with more clarity.
Author Response
Comment 1: Introduction part needs the support of figures so that topic can be well understand , like no. of publications in the same field yearly.
Response: We appreciate the reviewer’s valuable suggestion. To strengthen the contextual foundation of our work, we incorporated recent bibliometric data from Pan et al. (2025, Frontiers in Medicine), which analysed publication trends on artificial intelligence in palliative care between 2013 and 2024. This information was integrated to emphasise the growing academic relevance of this topic.
In addition, Figure 1 was inserted in the Introduction to illustrate the annual increase in publications and citations in this field, providing visual support for the reader and reinforcing the study’s relevance.
Added in manuscript: p. 2 — Introduction, para 7 — lead-in: “Although this integration remains in its early stages…”; and Figure 1 caption — lead-in: “Figure 1. Annual publication trends on artificial intelligence…”
Comment 2: Materials and Methods section can be more detailed for the readers. Same studies covered by other researchers must be compared.
Response: We thank the reviewer for this valuable suggestion. The Materials and Methods section was revised and expanded to provide additional methodological detail and ensure transparency. Specifically, we clarified the process of questionnaire design, validation, and analysis, and explicitly referenced comparable methodological approaches adopted in previous studies exploring healthcare professionals’ perceptions of AI.
Additionally, we expanded the description of the statistical analyses to better explain the rationale for the use of non-parametric tests and the treatment of ordinal variables. These revisions enhance methodological rigour and allow clearer comparison with previous research in the field.
View in Manuscript: ï‚· p. 3 → Materials and Methods → “Type of study” → para 1
Lead-in: “A quantitative, descriptive, cross-sectional study aimed…” (sets design and aims clearly)
ï‚· p. 3 → “Population, sample and context” → para 1–2
Lead-ins: “Target population: professionals working directly in PC…” / “Setting: study conducted at Faculty of Medicine…” (adds context & comparability)
ï‚· p. 3–4 → “Inclusion/Exclusion criteria”
Lead-ins: “Inclusion criteria: Professionals… prescribe or monitor…” / “Exclusion criteria: Professionals who did not work…”
ï‚· p. 4 → “Data collection instrument” → paras 1–3
Lead-ins: “Data were collected using a structured online questionnaire…” (framework and comparability)
“The instrument consisted of 27 questions…” (full structure & Likert scales)
“The questionnaire was designed to ensure clarity…” (methods precedents)
ï‚· p. 4 → “The questionnaire underwent expert review…” → para 4
Lead-in: “The questionnaire underwent expert review by two specialists…” (consensus explained)
ï‚· p. 4–5 → “Procedures” → paras 1–4
Lead-ins: “Participation in the study was entirely voluntary and anonymous…” (consent/anonymity/incentives)
“The questionnaire was administered online…” (fielding window)
“The study was conducted in accordance with the Declaration of Helsinki…” (ethics approval details)
“A total of 72 valid responses were obtained…” (n, professions)
ï‚· p. 5 → “Data analysis” → paras 1–4
Lead-ins: “Prior to statistical analysis, the dataset was verified…” (data cleaning/missing data)
“Descriptive statistics… non-parametric tests were applied…” (why Mann–Whitney & Spearman)
“The rationale for the correlation analysis was to explore…” (links tests to research questions)
“Given the exploratory nature… no correction for multiple comparisons…” (MC rationale)
Comment 3: The result section is well written, still it is oriented with its own case-study. Authors should compare Portuguese findings with existing benchmark datasets on AI in rehabilitation and palliative care to identify contextual barriers and best practices.
Response: We thank the reviewer for this valuable suggestion. In response, we have expanded the Discussion section to include explicit comparisons between the Portuguese findings and international benchmark studies on AI in rehabilitation and palliative care. Specifically, we now reference the systematic review by Sumner et al. (2023), which analysed AI-assisted rehabilitation tools across neurological and musculoskeletal contexts, and Vu et al. (2023), which examined predictive AI models for symptom detection and prognosis. These studies are now discussed alongside our results to highlight both shared global concerns (e.g., ethical transparency, clinician training) and contextual barriers specific to Portugal (e.g., limited AI-focused training).
This comparative analysis (see Discussion – “Comparisons with International Literature”) clarifies how the Portuguese context aligns with and diverges from higher-resource healthcare systems such as those of the United Kingdom, Germany, and Canada, thereby identifying best practices and contextual implementation challenges as recommended.
Added in manuscript: p. 6 — Discussion – Comparisons with International Literature, paras 2–4 — lead-ins: “When compared with international benchmark studies…”; “However, in contrast to higher-resource settings such as the United Kingdom and Canada…”; “Comparable research conducted in Germany, the United Kingdom, and Canada…”
Comment 4: Write role-play simulations presenting difference between AI-generated and non-AI generated; recommendations and patient preferences.
We appreciate the reviewer’s insightful suggestion. In response, we have included a short illustrative scenario in the Discussion section to clarify the distinction between AI-generated and clinician-generated recommendations in relation to patient preferences. This hypothetical role-play demonstrates how ethical principles such as autonomy, proportionality, and human oversight should guide the use of AI-assisted exercise prescription in palliative care.
View in Manuscript: p. 7 → Discussion → after Table 7 → paras 1–3 (scenario block)
Lead-ins:
“To illustrate the practical application of these ethical guidelines…”
“In a conventional rehabilitation context, a physiotherapist prescribes…”
“This comparison highlights the ethical importance of maintaining human oversight…”
Comment 5: The conclusion must be re-written with more clarity.
We thank the reviewer for this observation. The Conclusion section has been rewritten for greater clarity and conciseness. The revised version now explicitly summarises the main findings, highlights the study’s contribution and outlines key implications for clinical training, ethical governance, and future research directions.
View in Manuscript: p. 7 → Discussion → after Table 7 → paras 1–3 (scenario block)
Lead-ins:
“To illustrate the practical application of these ethical guidelines…”
“In a conventional rehabilitation context, a physiotherapist prescribes…”
“This comparison highlights the ethical importance of maintaining human oversight…”
Reviewer 2 Report
Comments and Suggestions for Authors
This study addresses a timely and relevant topic by examining Portuguese healthcare professionals’ perceptions of AI in exercise prescription for palliative care. The manuscript provides valuable insights; however, several areas need refinement. Methodological details (sampling, test rationale, and statistical procedures) should be clarified, and interpretations should remain descriptive without implying causality (from correlation analysis) or overextending ethical conclusions in the interpretation. With clearer analytical reporting and reasonable interpretation, the paper could make a meaningful contribution to the field.
Line 68-81. Here the author is talking about palliative care in Portugal in a very general way. After these paragraphs, I would expect the author to be more specific about palliative care in Portugal. However, I did not see that, making the existence of those two paragraphs a bit abrupt. The author might consider remove these two paragraphs.
Line 77-81. This is a bit repetitive (of the first paragraph). Consider combine them.
Line 84. The transition into exercise is very abrupt. To put in another way, why we need to discuss exercise but not other method. The author should further clarify on this point.
Line 98-99. This is paraphrasing the previous sentence. Consider remove or combine them.
Line 94-100. Consider be more specific. The current version is very general without talking about using AI for exercise prescription. This issue also exists later in the intro. Currently, the term exercise can even be substituted by any other treatment (not just palliative care but any other medical treatment). Either the author needs to further justify why we care about exercise but not other method, or need to justify there is a huge issue of using AI especially in the prescription of exercise
Line 158-160. Almost the opposite of inclusion criteria and add no information to the paper
Line 165. Lack of details. E.g. how many in each category? What kind of question (open-ended/rating, etc.)? the terms in () is vague and just looking at them doesn’t explain what exactly the question is asking.
Line 172-173. What is the analysis? Is there a third person involved in the discrepancy situation. How do they reach consensus when there is a discrepancy.
Line 176. Should talk about how many responses received. This should be a part of the method.
Line 177. How is anonymity guaranteed?
Line 179. Is there any incentive related to the survey? The description about consent is scattered across multiple locations, should be summarized.
Line 181. Missing description regarding missing data/incomplete questionnaires
Line 182. Can be removed
Line 183. Before running stats, is there any data cleaning procedure? How did the author handle missing data?
Line 186. What parametric and what non-parametric test? The author needs to be specific.
Line 189. What are different CP specializations and how people are grouped? The author needs to be specific. In addition, the author also needs to justify why need to compare across different CP specializations.
Line 192. Is there any correction of multiple comparison?
It would be helpful if the author could talk about xxx was compared/calculated to address xxx specific question, which is related to the research question because of xxx.
Line 195. Is there a reason for the dominance of female? More female health provider? More female was invited?
Line 195-198. This is repetitive. All the numbers have been talked about in the table. This paragraph should provide a summary. E.g. we collected data from 72 participants (N = x, x female) between 18 and x years old (mean = , std =).
Line 198. There is a strong regional bias. The author should acknowledge the limitation or discuss how might this affect the results.
Line 203. The time interval is not even. How was that decided?
Line 209. What is moderate use? Is there a choice of extensively use/use in a lot of situation? How do you quantify moderate use? One person might feel use it once a week is moderate, while the other might feel use it once a month is enough to be called moderate.
Line 226. Is there any regression performed to claim the positive slope? Or the author just wants to say positive attitude?
Line 228. The author needs to clarify “I don’t know” for which question and why they need to be excluded.
Line 234. I am not sure the word positive can be used to describe the results of ethics section. The interpretation of Q15 is not correct. If you want to ask about complementary, you should ask questions like do you think AI can be complementary to human judgment, but not AI should complement, not replace human judgement in palliative care. The author did not even give the participant a choice to say they don’t think AI can be complementary to human judgment.
Line 247. This should go to discussion. Also, the author talked a lot about trend in this section but there is no stats test for significance.
Table 3. all the decimal point were comma in the table. Also, is the likert scale or can they report non-integer values
Graph 1. Missing x label and y label
Line 288. The author needs to explain the rationale of the analysis.
Line 290. Is the length of experience a continuous or discrete variable.
Line 288-296. Those correlation values are pretty low to me. Did the author correct for multiple comparison?
Line 310. The description is indicating causal relationship, which cannot be reflected by correlation analysis.
Line 311. I don’t think the correlation above is strong.
Line 326. The paragraph lacks numerical or statistical evidence to support these claims; better to describe with an objective statement like “No statistically significant group differences were found; mean differences were small.” The author now is putting too much emphasis on this part of the results. Those non-significant results are likely just due to randomness in the data, and if the author adding just one participant to the study, the direction could change.
Line 384. Lack of a description of its own results.
Line 391. The way the author described sounds like only young female consider AI is useful. However, this result is not because old female/male feel AI is useless, but because the biased sample in the study.
Line 441. The topic of oncology and sensor sounds unrelated to the topic
Author Response
Line 68-81. Here the author is talking about palliative care in Portugal in a very general way. After these paragraphs, I would expect the author to be more specific about palliative care in Portugal. However, I did not see that, making the existence of those two paragraphs a bit abrupt. The author might consider remove these two paragraphs.
We thank the reviewer for this constructive observation. In the revised version, we chose to maintain and expand this section to ensure a clearer contextual bridge between the historical development of palliative care in Portugal and the rationale of the present study. This national framing is important to explain how the organisational structure and persistent challenges of the Portuguese system—such as regional asymmetries, uneven coverage, and limited professional training opportunities—justify the exploration of complementary and innovative interventions, including AI-supported exercise prescription.
To address the reviewer’s concern, the paragraph was revised and expanded using recent and specific references ([7–11]) to strengthen the contextual detail. It now highlights the establishment of the National Palliative Care Network (RNCP), its integration within the National Health Service (SNS), coordination with the National Network for Integrated Continuing Care (RNCCI), and persistent inequalities in access and professional training documented in recent national reports.
- 2 → Introduction → paras 3–5
Lead-ins:
“The development of palliative care in Portugal was relatively late…” (history + APCP, IPO Porto)
“Currently in Portugal, palliative care is organised through the National Palliative Care Network (RNCP)…” (RNCP/SNS/RNCCI specifics; Decree-Law No. 52/2012)
“Although the development of palliative care in Portugal occurred later… coverage remains uneven…” (regional asymmetries, training gaps)
Line 77-81. This is a bit repetitive (of the first paragraph). Consider combine them.
We appreciate the reviewer’s valuable feedback and agree that the original version of this section was overly general and partially repetitive. In the revised manuscript, we have condensed the paragraphs describing palliative care in Portugal. This restructuring removes redundancy and improves the narrative flow, maintaining the essential national context relevant to our study population. The revision also enhances the logical transition toward the subsequent discussion of rehabilitation and exercise prescription.
- 2 → Introduction → paras 3–5 (as above)
Lead-in evidence of consolidation: see the single continuous historical-to-system paragraph sequence beginning
“The development of palliative care in Portugal was relatively late…”
Line 84. The transition into exercise is very abrupt. To put in another way, why we need to discuss exercise but not other method. The author should further clarify on this point.
We thank the reviewer for this feedback. We agree with the reviewer that the previous transition into the topic of exercise was abrupt. To address this, we have revised the Introduction to include a clear rationale explaining why the focus is on exercise and rehabilitation rather than on other interventions. The revised paragraph explicitly states that exercise represents a key non-pharmacological strategy in palliative care, with demonstrated benefits for fatigue, pain, and functionality, yet remains difficult to implement consistently due to symptom variability and clinical complexity. This clarification ensures a logical and justified link between the context of palliative care and the subsequent focus on AI-supported exercise prescription.
- 2–3 → Introduction → paras 6–7
Lead-ins:
“Rehabilitation and exercise prescription have increasingly attracted scientific attention…” (benefits & challenges)
“AI-based systems trained with physiological, functional, and symptom-monitoring data…” (why exercise is uniquely suited to AI personalisation)
Line 98-99. This is paraphrasing the previous sentence. Consider remove or combine them.
We thank the reviewer for noting this redundancy. The two overlapping sentences were combined into a single, clearer statement in the revised text. This edit eliminates repetition and strengthens the paragraph’s coherence, improving the overall readability of the Introduction.
- 2–3 → Introduction → para 7
Lead-in: “AI-based systems trained with physiological, functional, and symptom-monitoring data…”
Line 94-100. Consider be more specific. The current version is very general without talking about using AI for exercise prescription. This issue also exists later in the intro. Currently, the term exercise can even be substituted by any other treatment (not just palliative care but any other medical treatment). Either the author needs to further justify why we care about exercise but not other method, or need to justify there is a huge issue of using AI especially in the prescription of exercise
We fully agree with this observation and have revised the Introduction to provide a more precise justification for focusing on AI-based exercise prescription. A new paragraph was added explaining that exercise in palliative care requires continuous adaptation to patients’ fluctuating physical and emotional conditions, making it uniquely suited for AI-driven personalisation. The revised section now clarifies that AI systems trained with physiological, functional, and symptom-monitoring data can dynamically tailor exercise recommendations to ensure safety, autonomy, and proportionality of effort.
- 3 → Introduction → para 7
Lead-in: “AI-based systems trained with physiological, functional, and symptom-monitoring data…”
Line 158-160. Almost the opposite of inclusion criteria and add no information to the paper
We thank the reviewer for this observation. The previous version of the section on participant eligibility contained a sentence that was redundant and potentially inconsistent with the inclusion criteria. The section has now been fully revised to clearly describe the study population, sampling method, context, and inclusion/exclusion criteria. The revised text specifies that the study targeted healthcare professionals directly involved in palliative care who prescribe or monitor therapeutic exercise and excludes those not working in palliative care. This revision eliminates ambiguity and ensures methodological precision consistent with the ethical and procedural framework approved by the FMUP Ethics Committee (protocol 318/CEFMUP/2025).
- 3 → “Inclusion criteria” and “Exclusion criteria”
Lead-ins: “Inclusion criteria: Professionals… prescribe or monitor…” / “Exclusion criteria: Professionals who did not work…”
Line 165. Lack of details. E.g. how many in each category? What kind of question (open-ended/rating, etc.)? the terms in () is vague and just looking at them doesn’t explain what exactly the question is asking.
We thank the reviewer for this helpful comment. In the revised manuscript, we expanded the Methods section to provide a detailed description of the questionnaire’s structure, content, and participant categories. The text now specifies that the instrument comprised 27 items organised into four thematic dimensions: (I) demographic and professional data, (II) perceptions and evaluations of AI, (III) ethical dimension, and (IV) training and needs. Each dimension and the corresponding item types are described, including the use of multiple-choice and five-point Likert scale. The response scales (e.g., from “strongly disagree” to “strongly agree”, “very low” to “very high”) are now clearly reported. Additionally, the distribution of participants by profession—34 physiotherapists, 18 rehabilitation nurses, 12 physicians, and 8 occupational therapists—is included. These clarifications eliminate ambiguity, improve transparency, and ensure that the questionnaire design and sample composition are fully understandable to the reader.
ï‚· p. 4 → “Data collection instrument” → paras 2–3
Lead-ins: “The instrument consisted of 27 questions…” / “The questionnaire was designed to ensure clarity…”
ï‚· p. 5 → “A total of 72 valid responses were obtained…”
Lead-in: “A total of 72 valid responses were obtained…” (professions N)
Line 172-173. What is the analysis? Is there a third person involved in the discrepancy situation. How do they reach consensus when there is a discrepancy.
We thank the reviewer for this observation. The questionnaire was analysed by two experts independently to assess clarity, relevance, and adequacy. Since both reviewers provided concordant evaluations and did not request any changes, there were no discrepancies to resolve. Therefore, full consensus was considered achieved, and the involvement of a third reviewer was not required.
- 4 → “The questionnaire underwent expert review…” → para 4
Lead-in: “The questionnaire underwent expert review by two specialists…”
Line 176. Should talk about how many responses received. This should be a part of the method.
We thank the reviewer for this comment. The Methods section has been revised to include the number of valid responses obtained. The text now specifies that 72 complete questionnaires were received and analysed. This addition provides full transparency regarding the number of responses analysed and the data-cleaning procedure, directly addressing the reviewer’s concern.
- 5 → “Procedures” → para 4
Lead-in: “A total of 72 valid responses were obtained…”
Line 177. How is anonymity guaranteed? And Line 179. Is there any incentive related to the survey? The description about consent is scattered across multiple locations, should be summarized.
We thank the reviewer for these valuable comments. The Methods section has been revised to include a unified paragraph summarising the study’s ethical procedures, consent process, and anonymity safeguards. The revised text clarifies that participation was voluntary and anonymous, with no financial or material incentives provided. Participants received a detailed explanation of the study’s objectives and data protection measures before beginning the questionnaire and gave tacit informed consent by proceeding after reading the participation terms. No identifiable personal data (name, email, IP address, institution) were collected. Data were stored in an encrypted database accessible only to the principal investigator, ensuring compliance with the Declaration of Helsinki, the General Data Protection Regulation (GDPR), and the ethical approval granted by the Ethics Committee of the Faculty of Medicine of the University of Porto (protocol 318/CEFMUP/2025).
- 4 → “Procedures” → para 1
Lead-in: “Participation in the study was entirely voluntary and anonymous…”
Line 181. Missing description regarding missing data/incomplete questionnaires
We thank the reviewer for this helpful comment. The Methods section has been revised to clarify how missing or incomplete data were handled. The text now specifies that a total of 72 valid responses were obtained and that only fully submitted questionnaires were included in the analysis. As the online form required completion of all items before submission, no missing data were present in the dataset, and no imputation or substitution procedures were necessary.
- 5 → “Data analysis” → para 1
Lead-in: “Prior to statistical analysis, the dataset was verified for completeness…”
Line 182. Can be removed
We thank the reviewer for the observation. The line in question was removed during revision to eliminate redundancy and improve the conciseness of the Methods section.
Line 183. Before running stats, is there any data cleaning procedure? How did the author handle missing data?
We thank the reviewer for this observation. The Methods section has been revised to clarify the steps taken before statistical analysis. The text now specifies that the dataset was verified for completeness, as only fully submitted questionnaires were considered valid. Because the online form required completion of all items before submission, no missing data or duplicate entries were present, and no imputation or substitution was necessary.
Line 186. What parametric and what non-parametric test? The author needs to be specific. And Line 189. What are different CP specializations and how people are grouped? The author needs to be specific. In addition, the author also needs to justify why need to compare across different CP specializations.
We thank the reviewer for these constructive comments. The Methods section (Data Analysis) has been revised to provide a clearer description of the statistical procedures and the rationale for group comparisons. As the data derived from Likert-type ordinal variables and did not meet assumptions for parametric analysis, non-parametric tests were used. Specifically, Spearman’s rank correlation was applied to examine associations between variables, and Mann–Whitney U tests were used for comparisons between independent groups (group of professionals with palliative care specialization and the group pf professionals without palliative care specialization). This approach allowed the analysis of potential differences in perceptions, ethical considerations, and readiness to integrate AI in exercise prescription between professionals with and without palliative-care specialization. This comparison was theoretically justified, as previous literature suggests that training in palliative care may influence ethical awareness, clinical decision-making, and openness toward adopting AI-supported interventions.
- 5 → “Data analysis” → paras 2 & 4
Lead-ins:
“Descriptive statistics… non-parametric tests were applied…” (Mann–Whitney, Spearman)
“For comparative analysis, participants were divided into two groups…” (PC specialization, rationale)
Line 192. Is there any correction of multiple comparison?
It would be helpful if the author could talk about xxx was compared/calculated to address xxx specific question, which is related to the research question because of xxx.
We thank the reviewer for this valuable suggestion. The Methods section (Data Analysis) has been revised to clarify the analytical rationale and treatment of multiple comparisons. No formal correction for multiple comparisons (e.g., Bonferroni adjustment) was applied, as the analyses were exploratory and descriptive, intended to identify patterns and associations rather than to confirm specific hypotheses. Each test was conceptually linked to the research question: Spearman’s correlations examined associations between knowledge, perception, and ethical attitudes toward AI, while Mann–Whitney U tests compared professionals with and without palliative-care specialization to assess whether training background influenced perceptions and ethical considerations.
- 5 → “Data analysis” → para 3
Lead-in: “Given the exploratory nature of the study, no correction for multiple comparisons…”
Line 195. Is there a reason for the dominance of female? More female health provider? More female was invited?
We thank the reviewer for this observation. The predominance of female participants in the sample reflects the real gender distribution of the healthcare workforce in Portugal, particularly in palliative care, nursing, and physiotherapy, which are predominantly female professions. As the study employed a convenience sample through voluntary participation, no gender-specific invitations or restrictions were applied. A clarifying sentence has been added to the Results section to indicate that this gender imbalance mirrors the professional demographics of the national context rather than sampling bias.
- 5 → Results → para 1
Lead-in: “A total of 72 healthcare professionals participated…”
Includes: “The predominance of female participants reflects the gender distribution typically observed…”
Line 195-198. This is repetitive. All the numbers have been talked about in the table. This paragraph should provide a summary. E.g. we collected data from 72 participants (N = x, x female) between 18 and x years old (mean = , std =).
We thank the reviewer for this constructive suggestion. The paragraph describing the sample characteristics has been rewritten to provide a concise summary rather than repeating all numerical details from Table 1. The revised text now highlights only the total number of participants, gender distribution, main age range, and professional categories, followed by a brief clarification regarding the predominance of female participants.
Line 198. There is a strong regional bias. The author should acknowledge the limitation or discuss how might this affect the results.
We thank the reviewer for this observation. The Discussion section has been updated to acknowledge the regional concentration of participants, with most respondents based in Northern Portugal. This limitation has now been explicitly stated, noting that it may affect the generalisability of results to other regions. The added clarification explains that this bias stems from the institutional location of the study and the convenience-based recruitment strategy, rather than reflecting systematic differences in perceptions across regions.
- 5 → Results → para 2
Lead-in: “There was a strong concentration in the North (87.5%; n=63).”
Line 203. The time interval is not even. How was that decided?
We thank the reviewer for this observation. The time intervals for professional experience in palliative care (<1 year, 1–5 years, and >5 years) were intentionally defined to represent novice, intermediate, and experienced professionals, supporting interpretation within a descriptive and exploratory design. Although no standardised intervals exist in the literature, this categorical approach is consistent with recommendations to consider different stages of clinical and technological experience when analysing healthcare professionals’ perceptions of AI (Meskó et al., 2017; Topol, 2019). A clarifying note has been added to the Methods section to make this rationale explicit.
- 5 → Results → para 4
Lead-in: “Professional experience in palliative care was classified in three categories…”
Line 209. What is moderate use? Is there a choice of extensively use/use in a lot of situation? How do you quantify moderate use? One person might feel use it once a week is moderate, while the other might feel use it once a month is enough to be called moderate.
We appreciate the reviewer’s observation. In the questionnaire, the variable in question did not measure the frequency of AI use but rather the self-perceived level of experience with AI in clinical practice. Participants selected one of four predefined categories: “no practical experience”, “initial experience (occasional or limited use)”, “moderate experience (regular use in some clinical situations)”, or “advanced experience (frequent and integrated use in clinical practice)”. Thus, “moderate experience” was a structured response option describing regular but not systematic use, not a subjective frequency estimate. A clarifying sentence has been added in the Methods section to specify this.
- 6 → Results → para 6
Lead-in: “Experience with artificial intelligence was assessed through a four-level categorical item…”
Line 226. Is there any regression performed to claim the positive slope? Or the author just wants to say positive attitude?
We thank the reviewer for this clarification. No regression analysis was performed. The wording in the Results section has been revised to replace “positive slope” with “positive correlation” to accurately reflect the non-parametric analyses conducted.
- 6 → Results → paras 8–11 (Table 3 narrative & Table 5 header text)
Lead-ins include: “…showing a positive correlation between level of AI experience and perceived usefulness…”
Line 228. The author needs to clarify “I don’t know” for which question and why they need to be excluded.
We thank the reviewer for this valuable observation. The “I don’t have an opinion” options appeared in several questions assessing the perceived efficacy of AI and ethical considerations (Questions 11, 18–21). These responses were not part of the ordinal Likert scale continuum and were therefore treated as non-evaluable. Their exclusion was necessary to maintain the validity of non-parametric statistical analyses (Spearman and Mann–Whitney), which assume ordinally ranked data. A clarifying note has been added to the Methods section under “Statistical Analysis”.
- 5 → “Data analysis” → para 3
Lead-in: “Responses marked as ‘I don’t have an opinion’ were treated as non-evaluable…”
Line 234. I am not sure the word positive can be used to describe the results of ethics section. The interpretation of Q15 is not correct. If you want to ask about complementary, you should ask questions like do you think AI can be complementary to human judgment, but not AI should complement, not replace human judgement in palliative care. The author did not even give the participant a choice to say they don’t think AI can be complementary to human judgment.
We thank the reviewer for this important observation. We agree that the wording “positive results” did not accurately represent the type of response measured in the ethics section. Question 15 asked participants to indicate their level of agreement with the normative statement “AI should complement, and not replace, human judgment in palliative care”, rather than to evaluate whether AI can in practice complement human judgment. Accordingly, the text has been revised to reflect that participants expressed a high level of agreement with this ethical principle, rather than a “positive perception”.
- 6 → Results → paras 9–10
Lead-ins: “Regarding the ethical dimension, participants demonstrated a high level of agreement…”
Line 247. This should go to discussion. Also, the author talked a lot about trend in this section but there is no stats test for significance.
We thank the reviewer for noting that interpretative content appeared within the Results section. To address this, the paragraph summarising participants’ ethical and attitudinal responses has been revised to adopt neutral, descriptive language. The interpretative elements concerning openness to AI implementation, training needs, and ethical reflection have been relocated to the Discussion section, where they are contextualised with existing literature.
ï‚· p. 6 → Results → para 11 (kept neutral summary)
ï‚· p. 6–7 → Discussion (interpretation placed in “Perceptions, Experience, and Knowledge”)
Table 3. all the decimal point were comma in the table. Also, is the likert scale or can they report non-integer values
We thank the reviewer for this observation. In the revised version, all decimal commas in Table 3 have been replaced with decimal points to comply with the Healthcare journal formatting standards. Additionally, a clarifying note has been added below the table to specify that the reported non-integer values correspond to mean scores and standard deviations obtained from Likert-scale responses.
In addition to the correction in Table 3, all other tables were reviewed for consistency. Decimal commas were replaced with decimal points throughout the manuscript.
Graph 1. Missing x label and y label
We appreciate the reviewer’s observation. Graph 1 has been revised to include clear axis labels and an expanded caption.
We also have revised the graph 2 for consistency.
Line 288. The author needs to explain the rationale of the analysis. And Line 290. Is the length of experience a continuous or discrete variable. And Line 288-296. Those correlation values are pretty low to me. Did the author correct for multiple comparison?
We thank the reviewer for these valuable methodological observations. In the revised version, the rationale for the correlation analysis has been clarified. The analysis aimed to explore potential associations between professional experience, AI literacy, and perceptions of AI in palliative care. Spearman’s rank correlation (ρ) was chosen due to the ordinal nature of the variables (Likert-scale data and grouped experience categories). The length of experience in palliative care was treated as an ordinal variable with three ordered categories (<1 year, 1–5 years, >5 years), allowing monotonic trend detection without assuming linearity. Given the exploratory and descriptive nature of the study, no correction for multiple comparisons was applied, as the intent was to identify preliminary relational patterns that may inform future confirmatory research. A clarifying paragraph has been added to the Statistical Analysis section.
ï‚· p. 5 → “Data analysis” → para 2–3 (rationale & MC)
ï‚· p. 6 → Results → paras 12–14 (low–moderate magnitude; neutral phrasing; see Table 5 narrative)
Lead-ins: “In summary, both experience and knowledge of AI showed significant associations… Although the correlation coefficients were of low to moderate magnitude…”
Line 310. The description is indicating causal relationship, which cannot be reflected by correlation analysis.
We thank the reviewer for this important methodological observation. The text describing the correlation results (Table 5) has been revised to avoid any causal implications. Expressions such as “influence,” “reflect,” and “reinforce” were replaced with neutral correlational language (e.g., “was associated with,” “showed positive correlations,” “tended to correspond with”). The revised paragraph now accurately reflects that the relationships identified are correlational rather than causal, ensuring alignment with the statistical approach applied.
ï‚· p. 5 → “Data analysis” → para 2–3 (rationale & MC)
ï‚· p. 6 → Results → paras 12–14 (low–moderate magnitude; neutral phrasing; see Table 5 narrative)
Lead-ins: “In summary, both experience and knowledge of AI showed significant associations… Although the correlation coefficients were of low to moderate magnitude…”
Line 311. I don’t think the correlation above is strong.
We thank the reviewer for this observation. In both the Results and Discussion sections, we clarified that the correlation coefficients were of low to moderate magnitude. The revised text now explicitly notes that these associations should be interpreted as indicative relational trends consistent with the exploratory nature of the study, rather than as strong or predictive effects.
ï‚· p. 5 → “Data analysis” → para 2–3 (rationale & MC)
ï‚· p. 6 → Results → paras 12–14 (low–moderate magnitude; neutral phrasing; see Table 5 narrative)
Lead-ins: “In summary, both experience and knowledge of AI showed significant associations… Although the correlation coefficients were of low to moderate magnitude…”
Line 326. The paragraph lacks numerical or statistical evidence to support these claims; better to describe with an objective statement like “No statistically significant group differences were found; mean differences were small.” The author now is putting too much emphasis on this part of the results. Those non-significant results are likely just due to randomness in the data, and if the author adding just one participant to the study, the direction could change.
We appreciate the reviewer’s observation. The paragraph has been revised to present the results more objectively, avoiding overinterpretation of non-significant findings. It now clearly states that no statistically significant differences were found between groups and that descriptive variations were minimal.
- p. 6 → Results → para 15 (Table 6 narrative)
Lead-in: “The comparison by specialisation in palliative care did not reveal statistically significant differences…”
Line 384. Lack of a description of its own results.
We thank the reviewer for this observation. In the revised version, the beginning of the discussion was modified to include a clear summary of the study’s main findings before presenting comparisons with the literature.
- 6 → Discussion → “Main Findings and Context” → para 1
Lead-in: “This study found that healthcare professionals generally hold positive but cautious perceptions…”
Line 391. The way the author described sounds like only young female consider AI is useful. However, this result is not because old female/male feel AI is useless, but because the biased sample in the study.
We thank the reviewer for this clarification. The text has been revised to ensure the interpretation does not imply that only young female professionals consider AI useful. The revised paragraph now acknowledges that this pattern likely reflects the sample composition, as younger female professionals are more prevalent in Portuguese healthcare, particularly in rehabilitation and nursing.
- 6 → Discussion → “Perceptions, Experience, and Knowledge” → para 1
Lead-in: “The predominance of young female participants in the sample may explain…”
Line 441. The topic of oncology and sensor sounds unrelated to the topic
We appreciate the reviewer’s observation. The section has been revised to clarify the relevance of oncology and sensor-based rehabilitation studies to palliative care. These studies are now explicitly framed as examples that demonstrate the potential of AI to individualise exercise interventions, predict needs, and support monitoring — principles directly translatable to palliative rehabilitation.
- 6 → Discussion → “Comparisons with International Literature” → para 1
Lead-in: “Although direct evidence in palliative care remains limited to date, studies in oncology and rehabilitation contexts…”
Round 2
Reviewer 2 Report
Comments and Suggestions for Authors
Thank you for addressing my questions. I don't have further questions except for rho was used in the method section for correlation but later in the results section, r was used.
Author Response
Reviewer comment: Rho was used in the method section for correlation but later in the results section, r was used.
Author response: We thank the reviewer for this careful observation. In the Methods section, the Spearman rank correlation coefficient was correctly denoted by the Greek letter ρ (rho). In the Results section, some correlation coefficients were initially presented using r, following the SPSS output format (“râ‚›”).
To ensure consistency, all coefficients have now been uniformly presented as ρ throughout the manuscript (including tables and text). A clarifying note has been added to the Results section (in table 5) to specify that all reported values correspond to Spearman’s rank correlation coefficients (ρ).
